# Microbiological Contamination of Medicinal Products —Is It a Significant Problem?

**DOI:** 10.3390/ph18070946

**Published:** 2025-06-23

**Authors:** Stefan Tyski, Magdalena Burza, Agnieszka Ewa Laudy

**Affiliations:** 1Department of Pharmaceutical Microbiology and Laboratory Diagnostic, National Medicines Institute, 00-725 Warsaw, Poland; s.tyski@nil.gov.pl (S.T.);; 2Department of Pharmaceutical Microbiology and Bioanalysis, Medical University of Warsaw, 02-097 Warsaw, Poland

**Keywords:** microbial contamination of drugs, European Pharmacopoeia, quality of medicines, sterile drugs, microbial purity of drugs

## Abstract

Medicinal products available on the market should be characterised by therapeutic efficacy, high quality, and safety for patients. They must either be sterile or comply with the appropriate pharmacopoeial microbiological purity requirements. Pharmacopoeial monographs related to microbiological tests of drug quality were also referenced. Despite stringent regulations governing pharmaceutical production, irregularities in the microbiological quality of drugs still occur. These are monitored by relevant agencies, which may order the recall of defective product batches from the market. However, in recent years, numerous cases of microbiological contamination in drugs and drug-related infections have been reported. Both isolated incidents and larger outbreaks or epidemics linked to contaminated medicines have been documented. Various microorganisms, including Gram-negative and Gram-positive bacteria, anaerobes, and yeast-like and mould fungi, have been identified in medicinal products or in patients affected by contaminated drugs. Ensuring the appropriate purity or sterility of pharmaceutical raw materials; maintaining cleanliness in the manufacturing environment, facilities, and equipment; and adhering to hygiene protocols and Good Manufacturing Practice regulations are essential for the production of safe and high-quality medicinal products. The aim of this study is to collect and compile information on the microbiological quality of drugs available on the market, with particular attention to identified irregularities, objectionable microorganisms isolated from medicinal products, and drug-related infections.

## 1. Introduction

Medicinal products are not only taken by healthy individuals in good condition but also by sick patients, often with chronic ailments and weakened immune systems. These products must demonstrate therapeutic effectiveness and maintain high quality to ensure the safety of those who use them.

Two critical aspects of improper microbiological quality in drugs serve as key parameters in evaluating product quality. These issues may lead to changes in the physicochemical properties of the product, potentially affecting its shelf-life and, in certain cases, causing harm or even life-threatening risks. Microbiological contamination of drugs results in the following effects:* A reduction in drug quality. This includes alterations in the pharmaceutical form (e.g., disintegration of tablets, agglomeration of powders and granules), organoleptic changes (e.g., deterioration in the appearance, smell, and taste of syrups, colour changes or stains on tablets), and breakdown of preservatives, which may enable the growth of pathogenic microorganisms (e.g., *Pseudomonas aeruginosa*). Additionally, microbial activity can lead to the degradation of active substances, such as the inactivation of penicillins by bacterial β-lactamases, the hydrolysis of aspirin in suspension by bacterial esterases, or the degradation of atropine in eye drops and steroids in ointments due to fungal contamination. The degradation of preservatives and the degradation of active substances are two distinct and harmful processes. A medicinal product may lose its preservatives due to microbial enzymes while retaining its therapeutic properties—for example, in the case of eye drops. Conversely, microorganisms may degrade the active substances in a drug without affecting the preservatives; this can happen in preparations such as antibiotic gels and ointments [1,2,3,4,5,6].* A reduction in patient safety. This includes drug-related infections caused by the presence of live pathogenic microorganisms in medicinal products and medical devices used for drug administration. When a contaminated drug is administered, microorganisms can grow, multiply, and release toxic substances in the patient’s body. Drug poisoning, on the other hand, is caused by cellular and extracellular factors—primarily toxins and enzymes—produced by microorganisms that were present in the raw materials and the finished product. In such cases, live microbial cells may no longer be present, so an infection does not occur [3,7,8,9,10,11].

The above aspect is thoroughly examined in this publication (Figure 1).

Not only is the presence of objectionable, pathogenic microorganisms particularly dangerous for patients taking contaminated drugs, but the extracellular factors released during microbial growth (such as toxins, and enzymes), and cell decay products (such as bacterial endotoxins—lipopolysaccharides) also play a significant role. It should be borne in mind that during the shelf life, microorganisms may die, but toxins (LPSs, enterotoxins, mycotoxins) may still retain their activity, causing drug poisoning (not drug-related infections). Therefore, such a sterile product may still be harmful. Pharmacopoeial compendia pay special attention to the content of pyrogens, especially bacterial endotoxins in parenteral preparations and mycotoxins, which is discussed further.

The first documented tragic case of microbial drug contamination occurred in 1902 when a vial of an anti-plague vaccine developed by Waldemar Mordecai Haffkine was contaminated with *Clostridium tetani*. On 30 October 1902, in Mulkowal, Punjab, India, 19 people who received a dose from a single bottle (53N) developed tetanus and died, while the 88 others inoculated with vaccine doses from different bottles remained healthy. The cause of contamination in the single bottle was never fully explained [12,13]. As a result, Haffkine was dismissed from his position as the director of the plague laboratory and placed on leave from the Indian Civil Service.

Over the years, sporadic infections caused by microbial contamination of medicines have occurred, along with large-scale outbreaks. In 1965, an epidemic of typhoid fever in Sweden affected more than 200 people after they received the oral drug Thyroidinum, which contained contaminated raw material—thyroid gland powder from domestic animals. This outbreak was investigated and explained by Professor Kallings [14]. Between mid-1970 and April 1971, a nationwide epidemic of septicaemia in the U.S. was linked to contaminated intravenous products, resulting in an estimated 2000–8000 bloodstream infections caused by *Enterobacter cloacae* and *Pantoea agglomerans*. Approximately 10% of affected patients died during or shortly after becoming infected [15]. The Centers for Disease Control and Prevention (CDC) identified the contamination source: a new elastomer liner in the threaded caps of intravenous bottles manufactured by Abbott Laboratories. Bacteria contaminated the elastomeric inserts of the cup during the autoclave cooling process. The CDC found microbial contamination on the outer surface of the cap liner in 0–52% of unopened Abbott bottles from various batches [16,17,18]. Even in the 21st century, microbial contamination continues to pose serious risks. In 2012–2013, a large-scale fungal meningitis epidemic affected 751 patients across 20 U.S. states, resulting in 64 fatalities [19]. The outbreak was traced to the New England Compounding Center in Massachusetts, where epidural methylprednisolone acetate injections were contaminated with *Exserohilum rostratum*, an environmental mould [20]. This fungus can cause opportunistic infections, particularly in immunocompromised individuals. Two years later, the CDC reported eight cases of recurrent *E. rostratum* meningitis, with a median relapse time of 90 days. These findings influenced treatment recommendations and long-term care guidelines for affected individuals [21].

Drug-related infections due to microbial contamination should be distinguished from infections caused by skin microbiota introduced during injection, in which *Staphylococcus* species are the most common culprits [22,23,24]. Drug recalls play a vital role in quality assurance by removing defective products from the market. The impact of recall procedures on the pharmaceutical industry was recently discussed by Miglani et al. [25].

In Europe, national regulatory agencies and the European Directorate for the Quality of Medicines & HealthCare (EDQM) oversee the quality of medicinal products through the Official Medicines Control Laboratories (OMCL) Network. If a defective batch is identified, the European Medicines Agency (EMA) Defective Product Report must be completed [26].

In the U.S., the Food and Drug Administration (FDA) monitors drug recalls (Table 1). Microbiologically related recalls involving both sterile and non-sterile drugs constitute a significant group. Although most recalls concern a lack of sterility assurance, the comprehensive process that includes all procedures and activities necessary to manufacture a sterile product, there are also cases involving excessive levels of bacteria or fungi and contamination with pathogenic microorganisms.

Ahmed et al. [7] focused on fungal contamination in pharmaceuticals based on data from the PubMed, CDC, and FDA websites from 2003 to 2021, with particular attention to the public health risk posed by fungal-contaminated compounded pharmaceuticals. In 2013, 55 patients died of fungal meningitis caused by contaminated steroid injections containing methylprednisolone acetate. Later, in 2021, *Aspergillus penicillioides* contamination was reported in ChloraPrep preparations, which was linked to improper storage conditions that enabled fungal growth.

Ensuring proper environmental quality in production, material storage, and quality control is essential. Tidswell and El Azab [31] emphasised the high costs of microbiological testing, particularly the expenses associated with cleanrooms equipped with HEPA filters and the costs of the analysis itself. A lack of appropriate equipment and financial resources for testing may contribute to the presence of microbiologically unsuitable drugs on the market.

The aim of this study is to draw attention to the microbiological status of medicinal products on the market and the severity and frequency of microbial contamination, which can lead to drug disqualification, patient infections, or even death. This study also presents strategies for improving microbiological quality and reducing contamination-related issues.

## 2. Pharmacopoeial Requirements Concerning the Microbial Status of Drugs and Methods of Testing Their Microbiological Quality

All human and veterinary medicinal products on the market must comply with the relevant pharmacopoeial microbiological requirements [32]. Drugs are classified as either sterile or non-sterile, with non-sterile products required to meet specific microbiological purity criteria. Sterility is mandatory for preparations administered parenterally, mainly injections and infusions, into the eyes, and on serious wounds and burns, including some cutaneous drugs and those used during surgical procedures. Additionally, veterinary intra-mammary and intrauterine preparations must be sterile. Other pharmaceutical products, depending on their route of administration (oral, rectal, vaginal, oromucosal, gingival, cutaneous (on intact skin), nasal, auricular, transdermal, or for inhalation), must comply with microbiological purity standards (Figure 2).

The microbiological quality of non-sterile pharmaceutical preparations and substances for pharmaceutical use is determined based on pharmacopoeial monographs, which set acceptance criteria according to the route of administration [32,33]. Pharmacopoeial tests enable the quantitative enumeration of mesophilic bacteria (total aerobic microbial count—TAMC) and fungi (total yeast and mould count—TYMC) that grow in an aerobic environment within 1 g or 1 mL of the tested sample. These tests also allow for the determination of specified microorganisms under defined conditions.

To enhance the registration, production, and quality control of pharmaceuticals worldwide, pharmacopoeial standards are being unified through the harmonisation of monographs from the European Pharmacopoeia (Ph. Eur.), the United States Pharmacopeia (USP), and the Japanese Pharmacopoeia (JP). This harmonisation process has been ongoing for several years, coordinated by the International Council for Harmonisation of Technical Requirements for Pharmaceuticals for Human Use (ICH) and the International Cooperation on Harmonization of Technical Requirements for Registration of Veterinary Medicinal Products (VICH). Two key microbial pharmacopoeia monographs, covering sterility testing [34,35,36] and the examination of non-sterile products [32,33,37], have already been harmonised. Additional monographs have been developed to address quantitative microbial contamination testing in pharmaceuticals [38,39], with special attention given to oral medicines of natural origin, which often have a higher bioburden [40]. The following pharmacopoeial monographs, which outline qualitative methods for detecting microbial contamination and identifying undesirable microorganisms, have also been harmonised [41,42,43]. Recently, microbiological purity requirements for live biotherapeutic products (e.g., probiotics) have been introduced [44,45,46]. A separate area of concern is the microbiological testing of cell-based preparations, which should be sterile [47].

Microbiological testing of pharmaceuticals focuses on detecting and identifying bacteria and fungi. However, the detection of non-cellular biological pathogens, such as viruses and prions, is not included in these tests, despite their potential to contaminate medicinal products. Instead, pharmacopoeial compendia adopt a risk-reduction approach. The Ph. Eur. has developed two monographs addressing viral safety [48] and minimising the risk of transmitting animal spongiform encephalopathy agents (prions) through human and veterinary medicinal products [49].

In the case of parenteral pharmaceuticals, avoiding the presence of pyrogens is crucial. These fever-inducing substances, mainly endotoxins from Gram-negative bacteria (specifically the lipid A component of lipopolysaccharides—LPSs) and non-endotoxins (e.g., certain peptidoglycan structures from Gram-positive bacteria, enterotoxins, fungi, viruses, or specific chemical compounds), are not eliminated by standard sterilisation processes. When pyrogens enter the bloodstream as a parenteral pharmaceutical product contamination, they exhibit their biological activity and can pose risks to human health, ranging from mild fever to severe septic shock and even death. A number of pharmacopoeia monographs have been developed to outline tests for detecting the presence of pyrogens. The first widely used pyrogen test was based on evaluating the rectal temperature of rabbits before and after intravenous administration of the test preparation into the marginal vein of the ear [50,51,52]. Franco et al. [53] discuss the detection and evaluation of endotoxins from a pharmaceutical perspective. Due to the reduced use of laboratory animals, this in vivo test will be withdrawn from the Ph. Eur. as of 1 January 2026. In 1964, Levin and Bang [54,55] discovered the coagulation of Limulus amoebocyte lysate (LAL) in the presence of Gram-negative bacterial endotoxins. This in vitro test, which uses the haemolymph of the horseshoe crab, specifically detects endotoxins, a subset of pyrogenic substances [56]. The LAL test has been harmonised in relevant pharmacopoeia monographs [57,58,59]. However, to detect all pyrogens without the use of animals, the monocyte activation test was developed [60]. More recently, a new test utilising recombinant reagents, such as factor C, for bacterial endotoxin evaluation, has been introduced as an alternative to the traditional LAL test in both the Ph. Eur. and USP [61,62]. To ensure the apyrogenicity of parenteral drugs, the apyrogenicity of the primary packaging should be taken into account. A dedicated Ph. Eur. monograph has been developed [63]. The most commonly used depyrogenation process for glass and other high-temperature-resistant materials is dry heat, typically at a temperature of at least 250 °C for a minimum of 30 min. However, low-temperature procedures may also be applied, such as endotoxin removal by washing with water for injection (<0.25 IU endotoxin/mL) or chemical treatments using strongly alkalising, oxidising, or reducing agents in liquid or gaseous form, followed by rinsing with water for injection. The reduction in recovered endotoxin should be at least 3 log_10_.

In addition to commercially available ready-to-use medicines in their final packaging, some public and hospital pharmacies prepare compounded prescription medications, including parenteral nutrition solutions. A significant number of compounded medications are produced annually in the U.S. to meet patients’ unique needs [64]. Compounding allows for personalised dosing and formulations for patients who cannot use commercially available drugs due to allergies, rare diseases, or specific dosage requirements. To ensure patient safety and minimise risks such as contamination, infection, or incorrect dosing, the USP has developed a dedicated monograph [65] for compounded sterile preparations.

The issue of microbiological contamination in medicines is, to some extent, addressed in Directive 2004/23/EC of the European Parliament and of the Council of 31 March 2004, which establishes quality and safety standards for the donation, procurement, testing, processing, preservation, storage, and distribution of human tissues and cells. In accordance with this directive, the European Pharmacopoeia Commission has developed a Ph. Eur. monograph on the microbiological examination of human tissues [66]. For comprehensive information on infections in organ recipients and the severity of microbiological issues related to transplantation and the use of preservation fluid for organs prior to transplantation, Laudy and Tyski have prepared a relevant review [67].

## 3. Contamination of the Manufacturing Environment, Equipment, and Raw Materials

The quality of medicinal products manufactured in plants located in different geographic regions varies. However, it is undeniable that this quality has significantly improved since the introduction of Good Manufacturing Practice (GMP) regulations [68]. The recently amended Annex 1, “Manufacture of Sterile Medicinal Products”, is of particular importance [69]. Environmental parameters for pharmaceutical production are monitored, and their requirements are specified in GMP guidelines and relevant documents [70,71]. The USP has also emphasised the importance of microbiological control and monitoring of aseptic processing environments, including cleanrooms [72].

Improving abnormal production conditions, including reducing microbiological contamination of equipment and production areas, may incur substantial costs. For example, Baxter had to pay USD 18.2 million in a settlement over mould presence at a manufacturing facility [73]. Various sources of bioburden in pharmaceutical products have been identified [1]. Researchers have analysed microbiota in manufacturing sites, including cleanrooms, and common microorganisms present in raw materials, excipients, pharmaceutical dosage forms, and packaging materials. Inadequate cleaning and maintenance of production equipment and machinery can lead to the accumulation of microbial contaminants. Airborne microorganisms may be introduced into pharmaceutical products due to inadequate ventilation and improper handling. Additionally, manufacturing personnel can transfer bacteria and fungi to raw materials and finished products, particularly if hygiene protocols are not properly followed. The purity of the environment (air, surfaces, and personnel) for preparing parenteral medicinal preparations is critical, and such production must take place in rooms classified as Grades A-D for microbiological purity [69]. A meta-analysis comparing microbial contamination of parenteral drugs prepared in clinical versus pharmacy environments found higher contamination rates in clinical settings, ranging from 1.09% to 20.70%, whereas contamination rates in pharmacy environments were mostly 0.00%, with one study reporting 0.66% [74]. Recent guidance recommends that hospitals review reconstitution processes and implement risk-reducing measures to enhance patient safety in parenteral therapy. Microbiological monitoring of the environment in pharmaceutical facilities has shown the presence of staphylococci, bacilli, and micrococci [70]. Sandle [75] conducted a 9-year study reviewing more than 9000 microbial isolates from different cleanroom grades. The predominant microorganisms were Gram-positive cocci associated with human skin, although microorganisms from environmental sources, such as airborne and surface samples (Gram-positive rods), and water (Gram-negative rods), were also detected, though in lower numbers.

An extremely important problem is the fungal contamination in cleanroom and non-cleanroom environments [8]. Mould-contaminated surfaces, equipment, and objects are particularly difficult to disinfect and eradicate microorganisms. The presence of mould has been observed even in cleanroom HEPA filtration systems, frequently detecting *Cladosporium* spp., and *Penicillium* spp., *Aspergillus* spp., and *Paecilomyces* spp. [76]. There can be many sources of mould contamination in the pharmaceutical production environment. Moulds are common in nature, and mould spores are carried in the air. Moisture, on the other hand, favours mould growth. Any flooding of production plant surfaces, as a result of the failure of production cooling systems and municipal systems or floods, leads to mould contamination of the pharmaceutical production environment. A notable case occurred in October 2011 when Sanofi Pasteur’s sterile product manufacturing facility in Toronto, Canada, experienced flooding, which caused water damage to building materials and consequently fungal colonisation [71]. Environmental monitoring subsequently detected fungi, prompting the company to cease production in July 2012, repair the facility, and recall four tuberculosis Bacillus Calmette–Guérin strain (BCG) vaccine lots. This incident resulted in a global BCG vaccine shortage and significant financial losses. Studies on water-damaged buildings have compared airborne fungal counts inside and outside affected structures and analysed fungi from water-damaged materials such as walls and ceilings. In general, the ratio of main indoor/outdoor fungal counts includes *Cladosporium herbarum*, *Cladosporium cladosporioides*, and *Alternaria alternata* (50%); *Pencillium viridicatum* (26%); *Penicillium aurantiogriseum* (16%). The following fungal strains were isolated from visibly damaged wallboard: *P. aurantiogriseum*, *P. viridicatum*, *Paecilomyces variotii*, *Chaetomium globosum*, *Memnoniella echinata*, and *Stachybotrys chartarum*.

The microbiological quality of raw materials used in pharmaceutical manufacturing significantly impacts the level of microbiological contamination in the final product. Water, the most widely used substance in pharmaceutical production, is classified into two general types: bulk water (produced and used onsite) and packaged water (produced, packaged, and sterilised for extended microbial stability throughout their packaged shelf life) [77]. The quality of water used for pharmaceutical purposes is evaluated through pharmaceutical inspections, while the quality of drinking water is subject to sanitary regulations. Table 2 presents acceptance criteria for microbiological quality of non-sterile substances and pharmaceutical-grade water. Special guidance has been developed on the pharmaceutical use of different grades of water in the manufacture of active substances and medicinal products for human and veterinary use [78]. Pharmacopoeial requirements determine the physicochemical properties of various types of water and their microbiological status. Monographs have been developed concerning the quality of water purified [79], water for injections [80], and water for the preparation of extracts [81]. However, cases have been described where water for pharmaceutical purposes either does not meet the quantitative pharmacopoeial requirements or contains pathogenic bacteria. Microbiological control of water carried out in one pharmaceutical plant in Egypt showed that 13.2% of the tested samples did not meet the requirements for purified water, with Gram-negative bacteria such as *Stenotrophomonas maltophilia* and *Pseudomonas* spp. being isolated [70]. A study also investigated 16 samples of water used for pharmaceutical product preparation from different areas in a purified water system in Dhaka, Bangladesh, showing that although the total aerobic viable bacterial load in all tested samples was within the prescribed limit (<100 colony-forming units (CFU/1 mL), out of the 16 samples, 3 were contaminated with *Pseudomonas* spp. and 1 contained *Escherichia coli* [82]. The presence of non-culturable microorganisms in pharmaceutical waters and cleanrooms has also been reported [27].

In the harmonised text of pharmacopoeial monographs [32,33,37] concerning the microbiological quality of non-sterile pharmaceutical preparations and substances for pharmaceutical use, the acceptable limits are specified as TAMC—10^3^ CFU/g or CFU/mL and TYMC—10^2^ CFU/g or CFU/mL for non-sterile substances (raw materials) intended for pharmaceutical use. However, certain groups of medicinal products, such as aqueous preparations for oral use; oromucosal, gingival, cutaneous, nasal, and auricular preparations; preparations for vaginal or inhalation use; and transdermal patches require higher microbiological purity with limits of TAMC—10^2^ CFU/g or CFU/mL and TYMC—10^1^ CFU/g or CFU/mL. Although the active substance does not constitute 100% of the medicinal product, the difference in microbial limits between the raw material and the finished product, which is only one logarithmic unit, necessitates that manufacturers pay particular attention to preventing microbial growth in raw materials—especially the presence of specified microorganisms whose content in 1 g or 1 mL must be entirely excluded.

Microbial contamination of raw materials used in dry formulation manufacturing is often mitigated by drug production processes such as tablet compaction and granule drying. However, Cundell [83] reported pathogenic bacteria, including *Klebsiella aerogenes*, *Bacillus cereus*, *P. aeruginosa*, and *Staphylococcus aureus* and pathogenic fungi such as *Aspergillus flavus* and *C. albicans*, contaminating both coated and non-coated tablets in tropical and humid climates. Eissa and Mahmoud [84] assessed bacterial and fungal contamination in pharmaceutical raw materials according to USP standards [39,42] using a neutralisation assay. Their study successfully recovered low levels of microbial contamination, detecting strains such as *S. aureus*, *Bacillus subtilis*, *P. aeruginosa*, *C. albicans*, and *Aspergillus niger* in 12 raw materials, including cetostearyl alcohol, cetyl palmitate, glycerol, maize starch, and sorbitol 70%. In a review by Dao et al. [1], several pathogenic microorganisms such as *S. aureus*, *Serratia fonticola*, *Enterobacter* spp., *Proteus* spp., *Aspergillus* spp., and *Geotrichum* spp. were identified as contaminants in raw materials, like gelatine, lactose, Arabic gum, and starch. Obuekwe and Eichie [2] analysed the microbial purity of starch, lactose powders, and distilled water used in tablet formulation. Pathogenic strains of *S. aureus*, *Geotrichum* spp, *Aspergillus* spp., *Penicillium* spp., and *Rhodotorula* spp. were isolated. The authors examined the influence of *S. aureus*, *Geotrichum* spp., and *Aspergillus* spp. on tablet hardness and disintegration. Microbial deterioration of tablets was observed, which has serious pharmaceutical implications and reduces the therapeutic efficacy of drugs.

Eissa [3] evaluated microbiological risk and investigated bacterial contamination in non-sterile pharmaceutical raw materials and final products of various pharmaceutical forms in a single company in Cairo, Egypt, over the course of one year. The study showed that more than 60% of the investigated products were contaminated with *Micrococcaceae* (16.98% found in empty hard gelatine capsules), *Enterobacteriaceae* (18.86% found in vaginal cream applicators, plastic caps for bottles, sorbitol solution, finished hard gelatine capsule products, topical cream, and oral suspension), and *Bacillaceae* 24.53% (talc powder, liquid oral preparation, and finished hard gelatine capsule product). Gram-positive bacteria, such as *Micrococcus lylae* (11.3%), and Gram-negative bacteria, including *Bacillus circulans* (17%) and *Shigella* spp. (5.7%), represented 56.60% and 41.51%, respectively, of the total isolates from the investigated samples. Finished pharmaceutical products were contaminated with Gram-positive and Gram-negative bacteria in 53.33% and 68.18% of cases, respectively. This represents a very high percentage, indicating that such drugs pose a significant risk to the health of end users.

A separate issue is the microbiological purity of raw materials of natural origin, such as herbs and plants. Because of the potential for air and soil contamination, there is a significant risk that natural raw materials may contain large numbers of microorganisms, including pathogenic species. This issue is discussed further in the following sections.

In addition to raw materials of plant origin, the pharmacopoeia recommends the careful selection of raw materials of animal origin, particularly, collagen, gelatine, bovine blood and blood derivatives, tallow derivatives, animal charcoal, milk and milk derivatives, wool derivatives, amino acids, and peptones, to minimise the risk of prion contamination [49]. Prions can cause severe spongiform encephalopathies in humans, including Creutsfeldt–Jakob disease, kuru, Gerstmann–Sträussler–Scheinker syndrome, and fatal familial insomnia. Because prions cannot be detected in raw materials, various procedures are implemented to mitigate the risk of prion transmission [49].

Biopharmaceuticals (biological drugs) constitute a special category of medicinal products. They are primarily synthesised by eukaryotic cells grown in vitro, although therapeutic peptides such as insulin can be produced by prokaryotic cells, including bacteria. The active substance in a biosimilar drug, which is a highly similar copy of an innovative biological drug previously registered, is produced by living cells cultured in vitro. Both the materials and media for cell cultures and the biological production environment for peptides or proteins must be sterile and are subject to rigorous supervision and control. Mycoplasmas, small bacteria lacking cell walls, pose a significant threat to cell cultures in the pharmaceutical industry, affecting cell metabolism, proliferation, and chromosomal integrity. The detection of these bacteria is crucial for maintaining contamination-free cell cultures, which are fundamental to the manufacturing of biopharmaceuticals. The phenomenon of mycoplasma contamination (previously referred to as pleuropneumonia-like organisms) in HeLa cell culture systems was first recognised nearly 70 years ago [85]. To address this risk, a special Ph. Eur. monograph was developed for detecting mycoplasmas in pharmaceutical products [86]. The problem of mycoplasma contamination within the biopharmaceutical industry was highlighted by Armstrong et al. [87], who examined various new detection assays in comparison to pharmacopoeial methods for speeding up the identification of mycoplasmas in cell cultures and biological products. The problem of cell line purity, including those used in the production of biopreparations, is complex and is related to mycoplasma contamination and cross-contamination of cell lines with unrelated cells. It was detected that half of the cross-contaminated cell lines were found to be mycoplasma-contaminated [88].

Recombinant biopharmaceutical products are manufactured using living organisms, including Gram-negative bacteria that contain lipopolysaccharides. These endotoxins can be released into the lysate, where they may interact with and form bonds with biomolecules, including therapeutically active compounds. Therefore, manufacturing processes must be designed to minimise endotoxin contamination by monitoring raw materials and in-process intermediates at critical steps, in addition to final drug product release testing [89].

Takahashi et al. [90] highlighted the potential endotoxin contamination of single-use sterile surgical gloves. During surgical procedures, endotoxins can enter the patient’s bloodstream and cause health complications. Four types of gloves sold in Japan were tested by immersing them in saline and measuring endotoxin levels. Three of the four glove types showed endotoxin contamination. Additionally, an increase in cytokine production was observed in all contaminated gloves except for one, which contained anionic surfactants.

## 4. Contamination of Sterile Drugs

More than 50 years ago, Bühlmann [91] highlighted the necessity of microbiological control in the manufacture of sterile pharmaceutical products. Some data concerning microbial contamination of sterile preparations are listed in Table 3.

Among the drugs that must be sterile, ophthalmic drugs are the most commonly contaminated. According to pharmacopoeial requirements, all preparations used for the eyes must be sterile [101]. However, ophthalmic solutions, ointments, and gels used for diagnostic or therapeutic purposes have been found to be contaminated with microbial pathogens, leading to eye infections following their use [9,92,102]. Generally, single-use ophthalmic medicinal products do not pose a significant risk of contamination, except for contamination that may occur during application by the user. Therefore, these medicines do not contain preservatives. However, in the case of multi-dose ophthalmic drugs, contamination can occur on the packaging, cap, dosing system, or even within the drug itself due to contact with the user’s fingers during repeated use. A significant difference in contamination rates between drops used for one day versus those used for one week or one month was observed [95]. The contamination of the caps and the residual contents of 1-, 2-, 4-, and 7-day eye drop use in outpatient departments increased from 12% to 58% and from 34% to 58%, respectively. Therefore, multi-dose drug packaging often contains preservatives designed to inhibit microbial growth. However, some microorganisms can degrade these preservatives, allowing them to multiply within the drug and pose health risks to users. Taşli and Coşar [9] found that staphylococcal strains were the dominant contaminants in 34.4% of bottles containing benzalkonium chloride as a preservative. It should be noted that in some ophthalmology clinics, a single bottle of preserved eye drops is used for multiple patients to reduce hospital costs, increasing the risk of cross-contamination [103]. Somner et al. [104] conducted a study in the United Kingdom and observed that the risk of cross-contamination with coagulase-negative *Staphylococcus* species ranged from 1:400 to 1:80, depending on whether the bottle was reused once or six times. Another study showed that the prevalence of bacterial contamination of bottle tips and contents increased significantly as the duration of use extended beyond 1 week (*p* < 0.003) [93].

Numerous preparations are used for dry eye disease, a multifactorial ocular surface disorder affecting up to 50% of the population, depending on sex and ethnicity. As a result, tear preparations are frequently used.

Yilmaz et al. [105] conducted a prospective analysis of bacterial contamination in 410 multi-use topical ophthalmic drugs, including antibiotic eye ointments (n = 109), antibiotic drops (n = 103), steroid ointments (n = 67), and steroid drops (n = 131) used by 185 patients. Contamination was detected in 23 of 410 (5.6%) preparations, with the highest contamination rate found in steroid ointments (14.9%), followed by antibiotic ointments (5.5%). Eye drops had lower contamination rates, with steroid-containing drops at 3.1% and antibiotic-containing drops at 2.9%. Even drugs containing preservatives were contaminated, with *Staphylococcus* spp. being the dominant bacteria found on caps and in container contents. Users should take precautions to prevent contamination of ophthalmic medication containers. Daehn et al. [102] highlighted that the risk of contamination exists even when eye drops are used exclusively by healthcare professionals in a controlled operating room environment. Preservatives can cause ocular irritation, punctate keratitis, and allergic reactions, prompting the introduction of preservative-free artificial tears. However, preservative-free ophthalmic preparations are susceptible to microbial contamination once opened. Advanced age and fingertip contact were significant risk factors for microbial contamination (*p* < 0.05). These findings were confirmed by Lee et al. [106], who found that 9 of 20 (45%) single-use artificial tear vials subjected to fingertip contact were contaminated with *S. epidermidis* and *S. aureus*.

Research has also investigated the impact of packaging design on microbiological contamination. Da Costa et al. [107] evaluated the effect of instillation angle and nozzle tip geometry on cross-contamination risk in multi-dose ocular solution bottles. Adjusting the instillation angle to 90 degrees and using a nozzle design that prevents solution flow along the bottle’s side significantly reduced contamination rates. An analysis of the prevalence of microbial contamination in 140 multiple-user preserved eye drop containers in an outpatient clinic in Malaysia, taking into consideration the dropper tip and the residual drop contents in the bottle, showed that dropper tips were more contaminated than the residual contents. Coagulase-negative staphylococci were the most frequently isolated microorganisms (89%). The overall prevalence of contamination was 30%, with a significant difference in contamination rates between containers used after 14 days (19%) and those used after 30 days (11%) [103].

Microbiological contamination of multi-dose eye drops containing a fluorescent agent is also a significant problem, particularly when the drops are administered to multiple patients. Costa et al. [108] investigated microbial contamination of multi-dose fluorescein eye drops after 1, 4, and 8 days of use, comparing them to unopened bottles. The highest contamination was observed in eye drops exposed for just 1 day across all tested bottles. In this study, the contamination rate was high, with 55.5% of bottles found to be contaminated. The microbiological contamination profile revealed a prevalence of skin and conjunctival microbiota and environmental microorganisms, including *S. aureus* and coagulase-negative staphylococci. In another study [109], microbial contamination of multi-use bottles containing fluorescein sodium ophthalmic solution was evaluated. The bottles were collected from various eye hospitals in Ghana without specification of the duration of use. Contaminating microorganisms were detected in all 21 tested bottles, including clinically significant bacterial strains, as *Staphylococcus* spp., *Bacillus* spp., *Pseudomonas* spp., *Haemophilus* spp., and *Bordetella* spp. were isolated. Fungal contamination by *Aspergillus* spp., *Penicillium* spp., and *Cladosporium* spp. was also detected.

Besides ophthalmic drugs, reports indicate microbiological contamination in other sterile medicinal products. All parenteral medicinal products administered via intravenous, intra-arterial, intramuscular, subcutaneous, intra-articular, intraventricular, intrathecal, intracisternal, and intraocular routes must not only be sterile but also comply with the appropriate endotoxin limits as required by the specifications of pharmacopoeial standards. Cases of bacterial contamination in injectable drugs are rare but concerning. Watson et al. [110] reported a large outbreak of catheter-associated *Klebsiella oxytoca* and *E. cloacae* bloodstream infections among patients at an oncology chemotherapy centre linked to contaminated isotonic sodium chloride solution used for injection. Twenty-seven patients had blood cultures positive for *K. oxytoca*, *E. cloacae*, or both, and all had central venous catheters that had been flushed with the isotonic sodium chloride solution at the clinic. Pulsed-field gel electrophoresis confirmed that the *K. oxytoca* and *E. cloacae* isolates from the patients matched those obtained from multiple predrawn syringes and from the intravenous fluid and administration set in use at the clinic.

Some parenteral drugs are distributed in multi-dose containers and can be used over a prolonged period for one or more patients. A 2007 outbreak across nine U.S. states involved 162 cases of *S. marcescens* bloodstream infections caused by prefilled heparin and isotonic sodium chloride solution syringes [111,112]. Bacteria were found in unopened prefilled heparin and saline syringes manufactured by a specific company. Of the 83 *S. marcescens* blood isolates submitted for CDC investigation, 70 (84%) were genetically related to the *S. marcescens* isolates recovered from the prefilled syringes.

Ahmed et al. [7] compiled a critical review of outbreaks related to pharmacy compounding and drug recalls caused by fungal contamination. Injectable drugs, such as betamethasone, trimcinolone, methylprednisolone, and linezolid, were found to be contaminated with various fungi, including *Penicillium* spp., *Cladosporium* spp., *Aspergillus* spp., *A. alternate*, *E. rostratum*, and *Cochliobolus hawaiiensis*. There have been multiple cases of fungal infections resulting from contaminated epidural, paraspinal, or joint steroid injections. In 2012, the FDA identified 137 cases and 12 deaths due to an *Aspergillus fumigatus* meningitis outbreak across 10 U.S. states linked to contaminated methylprednisolone acetate from a single compounding pharmacy [113]. A large outbreak of *E. rostratum* meningitis associated with methylprednisolone injections was also reported [20,114,115]. During the CDC investigation, seven other fungal species, including human and plant pathogens, were detected from clinical materials of infected patients: *Cladosporium* spp. in three patients, *Aspergillus terreus* in three patients, and one strain each of *Aspergillus tubingensis*, *A. fumigatus*, *A. alternate*, *Paecilomyces niveus*, and *S. chartarum* [115].

Another significant concern and challenge for healthcare is microbiological contamination related to parenteral nutrition, which is essential for preventing malnutrition in patients unable to receive adequate nutrients via the oral or enteral route. This sterile preparation rich in nutrients is administered over a long period at room temperature, creating an ideal medium for microbial growth. Contamination of parenteral nutrition solutions can occur in several ways, including the use of components contaminated during manufacturing, inadequate aseptic technique during preparation of the solution, sterilisation failures, contamination of multi-dose lipid emulsions or dextrose solutions, contamination during storage, technical difficulties during administration, or through an ascending method of infusion [116]. The issue of infections associated with parenteral nutrition has two dimensions: microbiological contamination of the product itself, which appears to be less frequent, and contamination of the route of administration, including catheter-associated bloodstream infections, which occur more frequently. *E. cloacae* isolates were detected in refrigerated aliquots of parenteral nutrition solutions, in blood cultures from infected newborns, and in in-use parenteral nutrition solutions [10]. All isolated strains exhibited the same antibiotic susceptibility pattern and genomic DNA profile. Catheter-related bloodstream infections remain a significant risk for parenteral nutrition patients, with reported incidence rates ranging from 1.3% to 26.2% [117]. Marra et al. [100] documented a wide range of pathogens isolated from bloodstream infections in long-term parenteral nutrition patients (Table 3). However, *S. epidermidis* strains, likely originating from the skin, were the most frequently isolated pathogens in catheter-related bloodstream infections in patients receiving central parenteral nutrition [24]. Despite advances in sterile techniques, line management, and sepsis treatment, catheter-related bloodstream infections still remain a significant cause of morbidity and a potential cause of mortality in critical care patients.

Another important problem is pyrogens in parenteral products. Up to now, there is limited information on the presence of pyrogens in medicinal products. Fennrich et al. [118] provided historical and current insights into pyrogen detection in drugs, along with future perspectives. More recently, Janů [119] presented new approaches to addressing pyrogen contamination in parenteral preparations. The most commonly used raw material in pharmaceutical production is purified water, which is typically obtained from water intended for human consumption. Zhang et al. [120] highlighted the potential contamination of such water with endotoxins. A study conducted in Japan [121] assessed endotoxin contamination in samples from a plant extract library. Endotoxins were detected in 48% (n = 139) and 4% (n = 5) of field-collected and crude drug samples, respectively, with some samples containing endotoxin concentrations exceeding 5.0 EU/mL. Notably, LAL tests performed on three plant extracts (*Sanguisorba officinalis*, *Oenothera biennis*, and *Lythrum salicaria*) were affected by the presence of polyphenols in these plants.

In July 1998, 57 moderate-to-severe endotoxin-like reactions occurred in the western U.S. states following the administration of once-daily dosing regimens of gentamicin produced by Fujisawa USA, Inc. [122]. The FDA acknowledged the presence of endotoxin in the gentamicin preparation; however, the concentration was within USP limits when the drug was administered as labelled (every 8–12 h). In November 1998, Fujisawa voluntarily withdrew all unexpired preparations of parenteral gentamicin, and the FDA did not enforce a recall [123].

Tidswell [124] conducted a systematic analysis of potential endotoxin-related safety risks associated with parenteral drugs and medical devices. Between 2012 and 2021, seven drug recalls (0.45% of total recalls) were linked to endotoxin contamination.

Endotoxins not only pose risks to human health but can also alter the pharmacokinetic parameters of drugs. In rats pre-treated with lipopolysaccharide endotoxin isolated from *E. coli*, time-dependent effects on hepatic and/or intestinal microsomal cytochrome P450 isozymes were reported [125].

As emphasised in this chapter, microbiological contamination of sterile drugs primarily occurs in ophthalmic and injectable medicines, with parenteral nutrition solutions being particularly vulnerable.

## 5. Contamination of Non-Sterile Drugs

Microbiological contamination of non-sterile drugs may involve exceeding the allowable limits for bacterial (TAMC) and fungal (TYMC) counts in conventional drugs [32,33], and contamination in live biotherapeutic products, including the total aerobic microbial contamination count (TAMCC) and the total yeast and mould contamination count (TYMCC) [45,46]. Depending on the route of drug administration, the presence of specific bacterial and fungal groups may also be unacceptable. Some data concerning microbial contamination of non-sterile preparations are listed in Table 4.

Drugs containing water, such as aqueous preparations for oral use (e.g., syrups) or nasal and ear drops, are particularly susceptible to microbiological contamination because water serves as an essential medium for microbial growth and multiplication. In contrast, the risk of contamination in solid dosage forms, such as tablets, capsules, powders, or granules, is significantly lower and largely depends on storage temperature and water activity [11]. Water activity refers to the content of free water as opposed to crystalline or atomically bound water [132,133]. The lower the water activity, the less favourable the conditions for microbial growth.

The most commonly reported contaminants in non-sterile, water-based drug formulations are Gram-negative rods belonging to the *Burkholderia cepacia* complex (BCC) [128]. The predominant species within this group include *B. cepacia* (formerly *Pseudomonas cepacia*), *Burkholderia cenocepacia*, *Burkholderia stabilis*, *Burkholderia vietnamiensis*, *Burkholderia ambifaria*, *Burkholderia multivorans*, and *Burkholderia pyrrocinia* [134]. These strains often exhibit intrinsic resistance to various antibiotics, including beta-lactams, aminoglycosides, polymyxins, and fosfomycin [135]. As opportunistic pathogens commonly found in soil and water, they can cause life-threatening infections in immunocompromised patients, posing a particularly serious risk to individuals with lung diseases, especially those with cystic fibrosis. BCC contamination accounts for 20–30% of pharmaceutical recalls related to non-sterile, especially water-based, preparations [136]. Moreover, BCC strains have been identified in purified water, disinfectants, cosmetics, household items, and even medical ultrasound gels [135]. Studies have also demonstrated that BCC isolates can degrade parabens used as preservatives [5]. The presence of BCC strains in disinfectant containers was already recognised in the 20th century [137,138]. Bacteria were detected in both opened and unopened containers of a povidone-iodine solution used by medical staff to disinfect the tops of multi-dose vials containing dialysis fluid additives, peritoneal fluid administration set connectors, and peritoneal dialysis system ports, leading to patient infections. *B. cepacia* strains were isolated from the peritoneal fluid and blood of six paediatric patients in a Texas hospital. Similarly, an outbreak of *Burkholderia cepacia* infections was linked to internally contaminated commercial 0.5% chlorhexidine solution in a neonatal intensive care unit [138]. Additionally, benzalkonium chloride-based antiseptics have also been found to be contaminated with BCC [139].

A significant concern is the contamination of various types of gels, including medicinal gels and ultrasound gels, with BCC strains. Between January 2017 and March 2018, an outbreak of *B. cepacia* infections caused by contaminated analgesic gels occurred in a tertiary hospital in China [140]. Nine patients in the urology ward developed hospital-acquired urinary tract infections, with two also developing bloodstream infections due to *B. cepacia*. In Argentina, a nosocomial polyclonal bacteraemia outbreak caused by different BCC species was reported in a hospital in Rosario, where contaminated ultrasound gel was identified as the source [141]. BCC-contaminated ultrasound gel was also responsible for a nosocomial infection outbreak in two Canadian centres in 2004 [5]. Additionally, in Vellore, India, seven paediatric patients developed BCC bacteraemia, with environmental surveillance confirming contaminated ultrasound gel as the infection source [142]. These findings highlight the potential role of contaminated ultrasound gels in healthcare-associated infections, which should not be overlooked.

An extremely important source of contamination of medicinal products is the purified water used in their production. Contamination most often occurs within water systems in pharmaceutical factories. In 2016, the FDA and CDC identified a direct link between contaminated batches of oral liquid docusate sodium, produced by a pharmaceutical company in Florida, and a multistate BCC outbreak [143]. The CDC investigated more than 300 reports of positive BCC cultures and confirmed 108 cases among hospitalised patients across 16 hospitalised in 12 U.S. states. FDA testing revealed BCC contamination in 24 of 200 samples of the liquid docusate sodium product, which had been manufactured using the company’s purified water system [127].

Frequent contamination of water-based medicinal products, cosmetics, and production facility water systems with BCC strains has prompted increased FDA attention to this issue [144]. The FDA has warned drug manufacturers about the hazardous contamination risks posed by BCC in non-sterile, water-based drug products [145]. In response to BCC outbreaks, the USP introduced a new monograph in 2013 addressing microbiological testing of non-sterile products for BCC [43]. More recently, Duong et al. [146] developed a culture-independent nucleic acid diagnostic method for detecting and quantifying BCC contamination in aqueous pharmaceutical products. This method, validated as equivalent to ISO/TS 12869:2019 (which is dedicated to the detection and quantification of *Legionella* spp.), has been proposed as a valuable tool for BCC detection.

In addition to BCC isolates, other pathogenic microorganisms, including Gram-positive bacteria, Gram-negative rods, and fungi, can also contaminate medicinal products. Drug manufacturing, distribution, and dispensing should be conducted in accordance with GMP regulations. However, in pharmacies and healthcare facilities located in Ihiagwa Community, Owerri, Imo State, Nigeria, sealed containers of paracetamol, chloroquine, and metronidazole tablets were opened by personnel and distributed to people [130]. These opened containers were stored at room temperature, and microbial contamination was detected in 42 of 50 (84%) samples tested. This study highlights the critical issue of microbial contamination in unsealed drug containers and underscores the importance of stringent quality control measures in the wholesale and retail pharmaceutical sectors.

Fungal contamination remains a significant challenge in maintaining microbiological purity. An analysis of FDA recall notifications from 1990–1999 to 2000–2012 revealed that fungal contamination accounted for 5% of microbiological-related recalls in the first period and 21% in the second period, making it the second most common reason for microbiological recalls after water-associated Gram-negative bacteria [147]. Cheng et al. [148] reported on 12 patients who developed an intestinal infection due to *Rhizopus microspores*-contaminated Allopurinol tablets, with 5 patients dying of intestinal mucormycosis.

In Africa, malaria is a major cause of mortality and morbidity, particularly among children under 5 years old. A combination of artemether and lumefantrine has been approved as the primary drug for children with acute *Plasmodium* spp. infections. Given the physiological immaturity of young children, the medicines they receive must be of appropriate quality. A study evaluated 90 samples of paediatric artemether–lumefantrine dry powders and 90 samples of paracetamol syrups obtained from selected drug stores in Accra, Ghana, for microbiological quality [146]. Sixteen paracetamol syrup samples showed contamination exceeding USP limits. Additionally, *P. aeruginosa* and *Salmonella* spp. strains were isolated from some syrup and powder samples. Overall, 4.44% of the sampled paediatric artemether–lumefantrine dry powders and 25.56% of the paracetamol syrup samples were found to be non-compliant with USP microbial specifications [149]. It is necessary to tighten controls not only on the manufacturing process of medicinal products but also on their storage and distribution of medicines in pharmacies and over-the-counter medicine outlets.

Ensuring that pharmaceutical manufacturers adhere to good manufacturing, distribution, and storage practices is essential to preventing contamination. Drug packaging should also be carefully considered because poor-quality packaging can contribute to microbial contamination. A study by Khana et al. [150] on the microbiological quality of oral solid dosage forms showed that 3 of 18 tested samples exceeded TAMC and TYMC limits, which was linked to the lack of tightness in primary packaging. Relevant medicine regulatory authorities should conduct frequent inspections of manufacturing facilities and perform post-marketing surveillance to ensure compliance with GMP guidelines.

## 6. Natural Pharmaceutical Products

Raw materials and products of natural origin, particularly those derived from plants, are highly susceptible to microbiological contamination. Although bacterial and fungal spores are the predominant contaminants associated with medicinal plants, a wide variety of bacteria, fungi, and viruses can be present on or within plant materials. This contamination arises from environmental factors, such as air and soil pollution, and various stages of medicinal herb production. Some data concerning microbial contamination of natural pharmaceutical products and herbs are presented in Table 5.

Studies indicate that contamination levels often depend on the plant’s proximity to the soil surface during growth [157]. The high microbiological contamination levels of natural pharmaceutical products are reflected in special pharmacopoeial monographs, which permit higher microbial limits (TAMC, TYMC) than in conventional medicines [41,42,43]. Kneifel et al. [157] conducted a comprehensive review on microbial contamination of medicinal plants, compiling extensive data on the contamination of different herbal parts, including cortex, floss, folium, fructus, herba, pericarpium, and radix. The authors focused on quantitative rather than qualitative data.

The quality and safety of herbal drugs are affected not only by fungal contamination but also by the presence of mycotoxins. Aflatoxin, ochratoxin A, and fumonisin B are among the main mycotoxins detected in herbal medicines, while *Penicillium* spp. and *Aspergillus* spp. are the most frequently identified fungi [158]. Qin and Guo [159] also highlighted the issue of mould and mycotoxin contamination in plant-derived medicinal materials. Chen et al. [160] investigated the prevalence of fungi and mycotoxins in medicinal herbs, revealing significant contamination, primarily by *Aspergillus* spp. and *Penicillium* spp. Ochratoxin A was detected at high levels in *Codonopsis* radix and *Scutellariae* radix. Additionally, three strains of *Penicillium citrinum* isolated from *Ganoderma lucidum* were found to produce citrinin, which was detected in *Scutellariae* radix at a concentration of 53 µg/kg.

A critical issue in herbal medicine contamination is the microbial load in products used in traditional medicine. Many traditionally prepared herbal medicines are likely to harbour a diverse range of potentially pathogenic bacteria. In Africa, several traditional herbal products, such as *Hypoxis hemerocallidea* (African potato), *Bulbine natalensis* (rooiwortel), and *Sutherlandia frutescens* (kankerbos), are commonly used to treat various ailments. Govender et al. [161] assessed the microbial quality of herbal medicines sold in shops in the Nelson Mandela Metropole, Port Elizabeth, South Africa. Their findings indicated significant bacterial and fungal contamination, suggesting inadequate storage and poor hygienic practices during preparation. Some medicines contained methicillin-resistant *S. aureus* strains and *Bacillus diarrhoeal* enterotoxins, posing serious health risks.

Walusansa et al. [162] analysed 14 published studies and reviewed the prevalence and dynamics of clinically significant bacterial contaminants in herbal medicines sold in East Africa from 2000 to 2020. Their findings confirmed that the herbal medicine industry in this region poses significant health risks due to the presence of clinically relevant bacterial contaminants, particularly *E. coli*, *S. aureus*, *Salmonella* spp., *P. aeruginosa*, *K. pneumoniae*, *Shigella* spp., *Proteus* spp., *Serratia erwinia*, and *Enterobacter* spp. The presence of enteric bacterial contaminants suggests possible faecal pollution of herbal medicines across the region. In a valuable systematic review, Ahiabor et al. [153] compiled information from 50 publications (2000–2024) on microbial contamination of herbal medicines in different African regions, with half of the studies originating from Nigeria. All but one study reported diverse bacterial contaminants in analysed herbal samples. Fungal contaminants were reported in 35 (70%) publications. Unfortunately, the authors did not compare their findings with the microbiological purity requirements of herbal medicines set out in the pharmacopoeia publications. However, the study highlights that many of the isolated microorganisms may pose a significant risk to consumer safety.

Similar studies conducted worldwide confirm that microbiological contamination of raw materials and medicines of natural origin remains a significant issue. In Iraq, an analysis of six natural pharmaceutical products revealed that one product from India contained 1.4 × 10^3^ CFU/g of *E. coli*, while another from Iraq contained 3 × 10^3^ CFU/g of *Salmonella* sp. and 4.5 × 10^3^ CFU/g of *E. coli* [163]. In Saudi Arabia, more than 80% of the population reportedly uses traditional herbal medicine [164].

A review of these cases highlights the importance of GMP compliance in mitigating microbiological contamination in herbal medicines.

## 7. Other Aspects of Medicinal Product Contamination

Traditional pharmacopoeial sterility and microbiological purity tests are time-consuming, requiring material culturing, microbial isolation, and identification. A special pharmacopoeial monograph on alternative methods for the control of microbiological quality has been developed [165]. Some of these methods, based on physicochemical, instrumental, and genetic assays, significantly reduce testing time but require validation against pharmacopoeial standards.

Research centres continue to explore new alternative methods of microbiological control of medicinal products. De Brito and Lourenço [166] developed a chemometric-based method for microorganism identification in pharmaceutical products using Fourier transform infrared spectroscopy with attenuated total reflectance. The identification of several bacterial and *Candida* reference strains using this rapid method was found to be compatible with classical microbiological assays. The chemometric method for rapid identification of microbial contaminants in pharmaceutical products was validated with a sensitivity of 93.5%, a specificity of 83.3%, and a detection limit of 17–23 CFU/mL of the sample. Wu et al. [167] designed an *E. coli* detection method using antibody-conjugated quantum dots as immunofluorescence probes. The recovery rate for *E. coli* in clomipramine hydrochloride was only 73.6% when the bacterial count was 3.7 × 10^7^ CFU/mL. Although the process took only 4 h, the method was not widely adopted because of its low detection efficiency and inability to identify microorganisms other than *E. coli*. Advanced Fourier transform infrared-based analytical techniques have also been employed to correlate bacterial contamination in pharmaceutical products with environmental microorganisms in cleanroom settings, offering a rapid and accurate monitoring method [168].

In addition to ready-made drugs manufactured by pharmaceutical companies, compounded drugs are prepared on a small scale by hospital pharmacies for individual patients with specialised medical needs, posing a potential risk of microbial contamination [64]. Ready-made medicines available in public pharmacies are currently manufactured and tested in accordance with GMP regulations, ensuring higher quality standards than for compounded drugs. Civen et al. [169] reported on an outbreak of *S. marcescens* meningitis in four patients who received epidural injections of betamethasone compounded at a community pharmacy. A few years later, an outbreak of *S. marcescens* bloodstream infections was also described in patients receiving parenteral nutrition prepared by a compounding pharmacy between January and March 2011 [170]. Nineteen cases were identified, and nine patients died. Irregularities were found in the mixing, filtration, and sterility testing stages. *S. marcescens* isolates were recovered from a pharmacy tap, a mixing container, and an open container of amino acid powder. Held et al. [171] reported two life-threatening sepsis cases caused by *B. cepacia* from contaminated intravenous flush solutions prepared by a compounding pharmacy. Similarly, Gershman et al. [172] described an outbreak of *Pseudomonas fluorescens* bloodstream infections affecting 80 patients in six U.S. states between December 2004 and March 2006, following exposure to contaminated heparinised saline flush solutions, also prepared by a compounding pharmacy. Pulsed-field gel electrophoresis analysis confirmed that clinical isolates from 50 (98%) of 51 patients were genetically related to isolates from unopened syringes.

Not only bacteria but also fungi can cause infections related to compounded drugs. Five cases of *Exophiala dermatitidis* infection were associated with injectable steroids from a compounding pharmacy [96]. A large fungal meningitis epidemic involving 751 patients, 64 of whom died, occurred across 20 U.S. states between 2012 and 2013 [19]. The source of this outbreak was linked to contaminated lots of preservative-free methylprednisolone acetate that had been used for epidural anaesthesia steroid injections and was produced at a compounding centre in Massachusetts. Laboratory evidence confirmed the presence of *E. rostratum* in specimens from 153 patients.

The above information concerns the microbiological contamination of drugs approved for marketing. However, as Pullirsch et al. [173] have shown, such contaminations can also be found in counterfeit and unapproved drugs. In Canada, microbial contamination levels exceeded USP and Ph. Eur. limits in 23% of the tested illegal samples. Drugs for sexual enhancement, containing phosphodiesterase type 5 inhibitors, were particularly affected, most often contaminated with *Bacillus* spp. strains.

Another important issue is the degradation of p-hydroxybenzoic acid esters by *Pseudomonas beteli* and *Burkholderia latens* [174]. These chemical compounds, known as parabens, are commonly used as preservatives in multi-dose water-containing medicines to inhibit the growth of microorganisms that may enter the drug package during its use. The decomposition of paraben-preserved, multi-dose ophthalmic drugs compromises their antimicrobial protection, making these medications potentially hazardous to users and increasing the risk of infections.

If a manufacturer discovers after releasing a drug batch to the market that it does not meet specification requirements, including microbiological quality, it is obligated to voluntarily recall that lot. Such a situation occurred with Pfizer Inc., which issued a voluntary nationwide recall for two lots of Relpax (eletriptan hydrobromide) 40 mg tablets, used for the acute treatment of migraines, because of potential microbiological contamination of non-sterile products [175]. Patients taking a product potentially contaminated with *Pseudomonas* spp. and *Burkholderia* spp. strains are at risk of bacterial dissemination from the gut to the bloodstream, potentially leading to serious, life-threatening infections. More recently, a manufacturer voluntarily recalled an Atovaquone oral suspension, indicated for the prevention and treatment of *Pneumocystis jirovecii* pneumonia, because of contamination with *Bacillus cereus* [176].

## 8. Conclusions

The issue of microbiological contamination of drugs and drug-related infections is multifaceted, involving sterility failures, inadequate manufacturing practices, and contamination related to the route of drug administration. The preparation and compounding of drugs in hospital pharmacies, along with poor handling and repackaging, also pose a risk of microbiological contamination, particularly in non-sterile pharmaceutical products. A broad spectrum of bacteria and fungi has been implicated in pharmaceutical contamination, leading to serious health risks.

Countries with stringent regulatory frameworks and GMP-compliant manufacturing processes report fewer cases of drug contamination. Improvements in automation, reduced manual handling, and increased awareness among production staff have contributed to enhanced drug quality. Rigorous adherence to pharmacopoeial standards, frequent inspections by pharmaceutical regulators, and strict GMP enforcement are essential to ensuring the microbiological safety of medicinal products. Additionally, meticulous audits conducted by pharmaceutical inspectors to assess compliance with GMP regulations in manufacturing plants and market surveillance of available products are crucial in guaranteeing the quality and safety of medicinal products.

Returning to the question posed in the title, whether microbiological contamination of medicinal products is a significant problem, the complexity of the issue must be considered. When drug production adheres to GMP requirements and takes place under appropriate conditions, the risk of microbiological contamination is low. Using high-quality raw materials, particularly those of natural origin that meet pharmacopoeial standards, further reduces this risk. However, drug-related infections remain a concern, particularly during drug administration. Even if GMP principles are followed and products meet sterility or microbiological purity criteria, improper hygiene practices during drug administration can increase the risk of infection. These infections may not necessarily arise from microorganisms in the non-sterile drug itself but rather from those present on the skin or in the patient’s environment.

## Figures and Tables

**Figure 1 pharmaceuticals-18-00946-f001:**
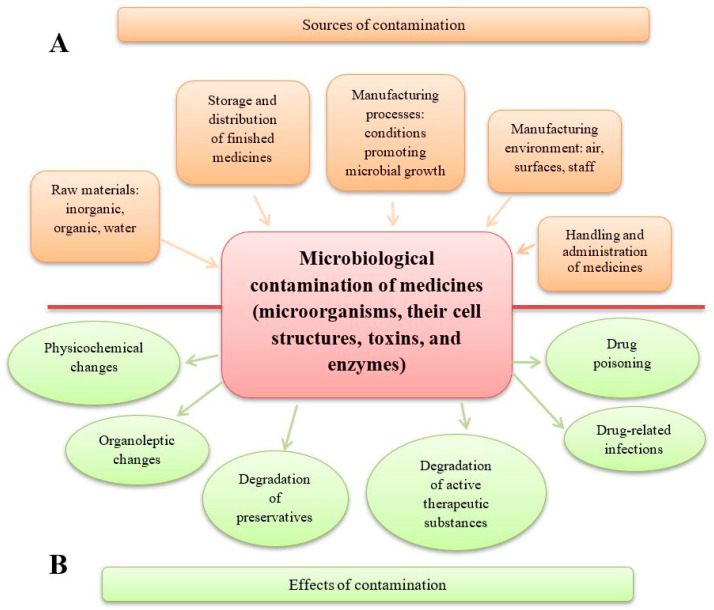
Sources and effects of microbiological contamination in medicinal products. (**A**) Sources of medicinal product contamination. A number of manufacturing stages are included, starting from the quality of raw materials to the production steps and the administration of drugs to patients. (**B**) The effects of medicinal product contamination may concern changes in the pharmaceutical form of drugs and loss of their properties, and adverse and harmful effects on the patient using the contaminated drug.

**Figure 2 pharmaceuticals-18-00946-f002:**
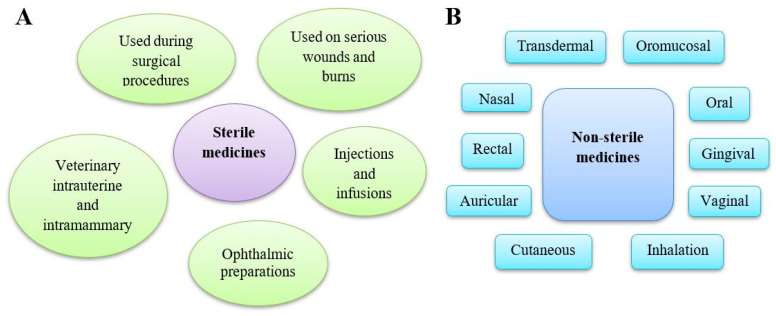
Types of medicinal products available on the market. (**A**) Sterile medicines are products for humans and animals free of microorganisms because they can pose a high risk of infection. Different categories of parenteral preparations may be distinguished: injections, infusions, concentrates or powders for injections or infusions gels for injection, implants, and intravitreal preparations. (**B**) Non-sterile medicines contain preparations used in various routes of administration, including cutaneous preparations used on intact skin. Several categories of preparation for inhalation may be distinguished: preparations to be converted into vapour, preparations for nebulisation, pressurised or non-pressurised metered dose preparations for inhalation, and inhalation powders.

**Table 1 pharmaceuticals-18-00946-t001:** FDA microbiologically related recalls from 1998 to 2023.

Years of FDA Analysis	Number of Microbiologically Related Recalls	Reason for Recalls	References
1998–2006	Total = 327; 134—non-sterile products; 193—sterile products.	Main contaminated microorganisms: *Burkholderia cepacia* (35 cases), *Pseudomonas* spp. (20 cases), *Salmonella* spp. (6 cases), *Ralstonia pickettii* (3 cases); 78% lack of sterility assurance.	[27]
2004–2011	Total = 642; 22% non-sterile; 78% sterile.	103 cases of objectionable organisms, including: *B. cepacia* (35 cases), *Pseudomonas* spp. (15 cases), yeast/mould (23 cases); 79% of sterile recalls—lack of sterility assurance.	[28]
2012–2019	713 non-sterile preparations; 1056—lack of sterility assurance.	Main contaminated microorganisms: *B. cepacia* (102 cases), *R. picketti* (45 cases), *Salmonella* spp. (28 cases), *Clostridioides difficile* (13 cases).	[29]
2012–2023	Total recalls = 15,710.	Sterility recalls (5766 cases), including lack of sterility assurance (2785 cases) and non-sterility (2621 cases); contamination (1712), including microbial contamination (686 cases). Main contaminated microorganisms: *B. cepacia Bacillus* spp., *Klebsiella* spp., *Candida albicans*, *Aspergillus* spp.	[30]

**Table 2 pharmaceuticals-18-00946-t002:** Acceptance criteria for microbiological quality of non-sterile substances and water for pharmaceutical use.

Raw Materials	Bacteria	Fungi	Endotoxins	References
Substances for pharmaceutical use	TAMC 10^3^ CFU/g or CFU/mL	TYMC 10^2^ CFU/g or CFU/mL	nt	[32]
Water purified	TAMC 10^2^ CFU/mL	nt	nt	[79]
Water for injections in bulk	TAMC 10 CFU/mL	nt	<0.25 IU/mL	[80]
Water for injections sterile	TAMC 0 CFU/mL	TYMC 0 CFU/mL	<0.25 IU/mL	[80]
Water for preparation of extracts	TAMC 10^2^ CFU/mL	nt	nt	[81]

CFU, colony forming unit; TAMC, total aerobic microbial count; TYMC, total yeast/mould count; IU, international unit of endotoxin; nt, not tested.

**Table 3 pharmaceuticals-18-00946-t003:** Microbial contamination of sterile preparations—examples of medicine types.

Medicine Types	Other Information	Isolated Microorganisms	References
Ophtalmic preparations	A total of 271 multi-use tear containers used by 168 patients were tested. Microbial contamination was detected in 33 (12.2%) of all containers, including ointments (32.0%), gels (11.7%), and drops (7.9%). Notably, collapsible tubes without preservatives had a significantly higher contamination rate (32%).	Contamination mostly by opportunistic bacterial and fungal strains, including *Pseudomonas stutzeri*, *P. aeruginosa*, *Bacillus licheniformis*, *Paenibacillus pabuli*, *Proteus mirabilis*, *P. agglomerans*, *Morganella morganii*, *Serratia marcescens*, and *Serratia liquefaciens*.	[4]
	A total of 92 eye drop bottles were examined: 43 bottles were opened and used for two weeks, and 49 bottles were unopened and sealed. The contamination rate was 34.8% in opened bottles and 10.2% in unopened bottles.	Six samples from opened bottles contain coagulase-negative staphylococci, and nine samples yielded one or two different microorganisms. Among unopened eye drop bottles, two samples contained *S. aureus*, two coagulase-negative staphylococci, and one *Bacillus* spp.	[9]
	Out of 123 multi-dose eye solutions, 10 were contaminated.	*P. mirabilis* was detected in 8 of 10 contaminated solutions.	[92]
	Out of 100 in-use eye drop vials, 11 were contaminated.	Staphylococci (7 strains), *Bacillus* spp. (2 strains), and single strains of *E. coli* and *Enterobacter* spp.	[93]
	In total, 242 eye drop vials were analysed, and bacterial contamination was detected in 5 vials.	3.9% coagulase-negative staphylococci and 1.0% *Acinetobacter* spp.	[94]
	A total of 200 eye drop bottles were tested after 1, 2, 4, and 7 days of use, and the contamination rates were as follows: 34%, 52%, 56%, and 58%.	Dominant contamination: *Staphylococcus epidermidis* (in 29.5% of tested containers), *Bacillus* spp. (16%), *Micrococcus* spp. (13.5%), *S. aureus* (8.5%), *Penicillium* spp. (15%), and *Aspergillus flavus* (8%).	[95]
Injection and infusion preparations	Two cases of meningitis infection due to contaminated injectable steroid (methylprednisolone) prepared by a compounding pharmacy.	*Exophiala* spp.	[96]
	In total, 8 metronidazole and 8 ciprofloxacin infusions preparations were tested; 2 of the metronidazole and 1 of the ciprofloxacin preparations were contaminated.	Microorganism identification was not performed.	[97]
	Fungal meningitis was caused by contaminated injections.	*Pithomyces chartarum*	[98]
Parenteral nutrition preparations	Eleven neonates were infected after parenteral nutrition	*E. cloacae*	[11]
	Multi-dose vials of medicinal salts, including potassium chloride, sodium chloride, and sodium bicarbonate, were examined. Bacterial contamination was identified in 36 of 637 (5.6%) vials.	Contamination by aerobic normal commensal microbiota. In Gram-positive bacteria (88.9%), *S. epidermidis* was the most common contaminant (44.4%). Gram-negative bacteria (11.1%).	[23]
	Catheter-related bloodstream infections in 850 patients who underwent central venous catheterisation for total parenteral nutrition. In total, 11.2% of patients in intensive care units and 12.1% of patients in non-intensive care units were infected.	Microorganism identification was not performed	[99]
	Bloodstream infections occurred in 37 of 47 long-term parenteral nutrition patients, and 23.8% of infection episodes were polymicrobial.	The most prevalent pathogen was coagulase-negative staphylococci (33.5%). Moreover, the following strains were also isolated: *Corynebacterium* spp., *S. aureus*, *Streptococcus* spp., *Leuconostoc* spp., *Lactobacillus* spp., *Bacillus* spp., *Propionibacterium* spp., *E. coli*, *Proteus* spp., *Acinetobacter baumannii*, *Serratia* spp., *P. aeruginosa*, *E. cloacae*, *Agrobacterium radiobacter*, *P. agglomerans*, *Citrobacter freundii*, *Acinetobacter lwoffi*, *Bacteriodes fragilis*, *Fusobacterium nucleatum*, *Ewingella americana*, *Kluyvera ascorbata*, *Candida tropicalis*, *Candida lusitaniae*, *Candida krusei*, *Rhodotorula rubra*, *Malassezia furfur*, and *Aureobasidium* spp.	[100]

**Table 4 pharmaceuticals-18-00946-t004:** Microbial contamination of non-sterile preparations—examples of medicine types.

Medicine Types	Other Information	Isolated Microorganisms	References
Nasal spray	CDC reported a manufacturer’s recall of over-the-counter oxymetazoline HCl 0.05% nasal spray due to bacterial contamination.	*Burkholderia cepacia* complex	[126]
	Molecular analysis confirmed a close genetic relationship between bacterial isolates from the manufacturer’s purified water, the liquid docusate sodium product, and patient clinical samples.	*B. cepacia* complex	[127]
Oral and topical preparations	Most of the 77 tested products met quantitative microbiological requirements of Ph. Eur.; 29 samples contained objectionable bacterial strains.	*B. circulans* (8 isolates) and single isolates of *Micrococcus luteus*, *Enterococcus faecium*, *P. agglomerans*, *R. pickettii*, *S. maltophilia*, and *Bordetella bronchiseptica.*	[128]
	All investigated products (mainly from India) were contaminated with microorganisms, with most exceeding the maximum acceptable counts. Syrups and suspensions were more contaminated than tablets and capsules.	*P. aeruginosa*, *S. epidermidis,* and *Klebsiella pneumoniae* were the most frequently isolated pathogens.	[129]
Tablets	Sealed containers of paracetamol, chloroquine, and metronidazole tablets were opened by personnel and distributed to people. These opened containers were stored at room temperature, and microbial contamination was detected in 42 of 50 (84%) samples tested.	*S. aureus*, *E. coli*, *P. aeruginosa*, *C. albicans*, and *A. niger*. The most common isolates were *S. aureus* (51.8% of bacterial isolates) and *C. albicans* (73.3% of fungal isolates).	[130]
Tablets, capsules, ointments, and syrups	In total, 1285 non-sterile pharmaceutical products manufactured by various pharmaceutical plants, before they were marketed, were tested, and 1.87% of the tested drugs were non-compliant with Ph. Eur. because of excessive microbial counts and the presence of pathogens excluded by Ph. Eur.	*Bacillus* spp., *Microccocus* spp., *Enterococcus* spp., *Aspergillus* spp., *Rhizopus* spp., *Alternaria* spp., and *Mucor* spp. were isolated.	[6]
	Among 10 non-sterile drugs, 5 contained between 10 and more than 1000 CFU/mL of pathogenic bacteria.	*Bacillus* spp. and *Klebsiella* spp., *Candida* spp. and *Aspergillus* spp.	[131]
Mouthwashes, syrups, skin creams, and other products.	Fungal contamination by moulds and yeasts was detected.	*Aspergillus* spp., *Fusarium* spp., *Rhizopus* spp., *Penicillium* spp., and *Candida* spp. were recorded most frequently.	[8]

CDC—Centers for Disease Control and Prevention; Ph. Eur.—European Pharmacopoeia.

**Table 5 pharmaceuticals-18-00946-t005:** Microbial contamination of natural pharmaceutical products and herbs—examples of medicine types.

Medicine Types	Other Information	Isolated Microorganisms	References
Herbal remedies	Widespread microbial contamination of 150 samples. The TAMC exceeded 5 × 10^7^ CFU/g in 59.33% of the samples, while 28% of the preparations had microbial counts below this threshold.	*S. aureus* in 65.33% samples, *E. coli* in 58.67% samples, *Salmonella typhi* in 46.67% samples, and *Shigella* spp. in 19.33% samples.	[151]
Herbal oral liquid preparations	The mean bacterial load of 50 analysed herbal medicine samples ranged from 0.0 CFU/mL to 2.9 × 10^12^ CFU/mL, while the mean fungal load ranged from 0.0 CFU/mL to 3.5 × 10^12^ CFU/mL. A total of 52% samples contained one bacterial contaminant each, 26% of the samples had two, while 20% had three contaminants. Four contaminants were recovered from one sample.	A total of 85 bacteria were recovered from 49 of the 50 samples, including 36 Gram-positive and 49 Gram-negative bacteria. Most often, it was *Bacillus* spp. (40%) and *Klebsiella* spp. (31.8%). Other contaminants: *E. coli*, *Staphylococcus* spp., *Salmonella* spp., and *P. aeruginosa*.	[152]
Herbal medicines	Data from 50 publications (2000–2024) on microbial contamination of herbal medicines in different African regions.	The most frequently isolated bacteria: *E. coli* (62%), *S. aureus* (60%), *Bacillus* spp. (54%), and *Pseudomonas* spp. (46%). The most frequently isolated fungi: *Aspergillus* spp. (40%), *Penicillium* spp. (28%), and *Candida* spp. (24%).	[153]
	From 47 products, including 18 creams, 15 liquids, and 14 powders, 58 bacterial strains were isolated, and all but 3 samples were contaminated with at least one microorganism. Most Gram-positive bacterial isolates were found to be multidrug-resistant.	Most commonly, *Klebsiella* spp., *Pseudomonas* spp., and *E. coli* were isolated.	[154]
	In total, 132 oral and topical products were tested. Bacterial and fungal contamination was detected in 51.5% and 35.6% of samples, respectively. A total of 31.8% of the herbal medicine samples exceeded the safety limits (≤10^5^ CFU/g), with 16.7% of homemade and 15.1% of commercial herbal medicines surpassing this threshold. Moreover, the tested water samples contained coliforms, rendering the water unfit for consumption.	The most commonly isolated were *S. aureus* (49.2%), *Salmonella* spp. (34.8%), *E. coli* (25.8%), and *P. aeruginosa* (14.4%).	[155]
Commonly used herbal medicines	A total of 173 fungal strains were isolated from 138 samples. Mycotoxin analysis revealed that *Fusarium* spp. primarily produced acetylated forms of deoxynivalenol, while *Alternaria* spp. mainly produced altertoxins.	The most frequently isolated were *Fusarium* spp. (28%) and *Alternaria* spp. (21%).	[156]

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
