# Peer review of "Microbiological Contamination of Medicinal Products —Is It a Significant Problem?"

_pharmaceuticals, 2025, doi:10.3390/ph18070946_

Round 1
Reviewer 1 Report
Comments and Suggestions for Authors
The manuscript presents a potentially valuable overview of microbiological contamination in medicinal products but suffers from significant technical shortcomings. The structure is disorganized, with lengthy, unfocused sections that lack thematic clarity. The text includes excessive historical examples without sufficient synthesis or critical analysis. Numerous grammatical errors and awkward phrasing reduce readability and hinder scientific communication. Data presentation is inconsistent, with poor integration of figures and tables; for instance, figures are referenced without meaningful explanation, and tables are presented with minimal interpretation. The manuscript also exhibits inconsistent citation formatting and several incomplete references. Overall, the technical execution does not meet the standards required for publication in its current form.
Comments on the Quality of English LanguageThe English could be improved to more clearly express the research.
Author Response
The manuscript presents a potentially valuable overview of microbiological contamination in medicinal products but suffers from significant technical shortcomings. The structure is disorganized, with lengthy, unfocused sections that lack thematic clarity. The text includes excessive historical examples without sufficient synthesis or critical analysis. Numerous grammatical errors and awkward phrasing reduce readability and hinder scientific communication. Data presentation is inconsistent, with poor integration of figures and tables; for instance, figures are referenced without meaningful explanation, and tables are presented with minimal interpretation. The manuscript also exhibits inconsistent citation formatting and several incomplete references. Overall, the technical execution does not meet the standards required for publication in its current form.
The text was corrected and improved according to the reviewers' suggestions. Figures were corrected and four new tables were added.
The English could be improved to more clearly express the research.
The manuscript has been submitted to Proof-Reading-Service.com from United Kingdom for editing and proofreading. The certificate is included.
Reviewer 2 Report
Comments and Suggestions for Authors
In this review, the authors discussed the topic “Microbiological Contamination of Medicinal Products – Is It a Significant Problem?”. After careful reading, I found the review to be scientifically meaningful and generally well written. However, the presentation is primarily text-based, with limited use of figures and tables. I recommend that the authors enhance the presentation style by incorporating more visual elements, such as tables and diagrams, to improve clarity and reader engagement.
Specific Comments:
Introduction (Lines 43–56):
Although the introduction is comprehensive, some parts, particularly lines 43 to 56, lack proper citation. Please ensure all statements are appropriately referenced to support the content.
Figure 1 – Source of Contamination:
In the section “Manufacturing environment: air, surfaces, staff and manufacturing processes: conditions promoting microbial growth,” please consider differentiating or integrating these elements more clearly to avoid redundancy.
Additionally, the term “finished medicines” needs clarification—do the authors mean expired medicines, or simply products ready for distribution?
Figure 1 – Effect of Contamination:
Consider revising the phrase to: “Degradation of preservatives and active substances” to improve clarity and avoid separating two closely related effects.
Also, please clarify the distinction between drug-related infections and drug poisoning—the difference is not clearly defined in the current version.
Clinical Outbreaks:
The major clinical outbreaks mentioned in the introduction would be more effective if presented in a tabular format.
Figure 2 – Classification of Medicines:
Please clarify whether cutaneous medicines fall under the category of non-sterile medicines.
Additional Figures and Tables:
I recommend that the authors include more figures and tables throughout the review to summarize key points, classifications, mechanisms, or case studies. This will enhance the overall readability and comprehension of the manuscript.
Author Response
In this review, the authors discussed the topic “Microbiological Contamination of Medicinal Products – Is It a Significant Problem?”. After careful reading, I found the review to be scientifically meaningful and generally well written. However, the presentation is primarily text-based, with limited use of figures and tables. I recommend that the authors enhance the presentation style by incorporating more visual elements, such as tables and diagrams, to improve clarity and reader engagement.
The text was corrected and improved according to the reviewers' suggestions. Figures were corrected and four new tables were added.
Specific Comments:
Introduction (Lines 43–56):
Although the introduction is comprehensive, some parts, particularly lines 43 to 56, lack proper citation. Please ensure all statements are appropriately referenced to support the content.
We incorporated appropriate citations.
Figure 1 – Source of Contamination:
In the section “Manufacturing environment: air, surfaces, staff and manufacturing processes: conditions promoting microbial growth,” please consider differentiating or integrating these elements more clearly to avoid redundancy.
The figure specifically lists three areas of the environment: air surfaces and staff, which are subject to microbiological control according to „The Rules Governing Medicinal Products in the European Union Volume 4 EU Guidelines for Good Manufacturing Practice for Medicinal Products for Human and Veterinary Use. Annex 1 Manufacture of Sterile Medicinal Products. Brussels,. EudraLex - Volume 4 - Good Manufacturing Practice (GMP). [https://health.ec.europa.eu/document/download/e05af55b-38e9-42bf-8495-194bbf0b9262_en?filename=20220825_gmp-an1_en_0.pdf] .
Additionally, the term “finished medicines” needs clarification—do the authors mean expired medicines, or simply products ready for distribution?
Finished medicines, (term used in pharmacy) mean products ready for distribution in the Fig 1 we write: “Storage and distribution of finished product” – we think it is understandable.
The Rules Governing Medicinal Products in the European Union Volume 4 EU Guidelines for Good Manufacturing Practice for Medicinal Products for Human and Veterinary Use, e.g. paragraph 8.34 “Where possible, finished product should be terminally sterilised, using a validated and controlled sterilisation process, as this provides a greater assurance of sterility than a validated and controlled sterile filtration process and/or aseptic processing.”
Figure 1 – Effect of Contamination:
Consider revising the phrase to: “Degradation of preservatives and active substances” to improve clarity and avoid separating two closely related effects.
Degradation of preservatives and degradation of active substances are two distinct and harmful processes. A medicinal product may lose its preservatives due to microbial enzymes while retaining its therapeutic properties—for example, in the case of eye drops. Conversely, microorganisms may degrade the active substances in a drug without affecting the preservatives; this can happen in preparations such as antibiotic gels and ointments.
We have combined the terms "Degradation of active substances" and "Reduction or loss of therapeutic properties" into "Degradation of active therapeutic substances".
Also, please clarify the distinction between drug-related infections and drug poisoning—the difference is not clearly defined in the current version.
Drug-related infections are caused by the presence of live microorganisms which, when a contaminated drug is administered, can grow and multiply in the patient receiving the medication.
Drug poisoning, on the other hand, is caused by cellular and extracellular factors—primarily toxins and enzymes—produced by microorganisms that were present in the raw materials and the finished preparation. In such cases, live microbial cells may no longer be present, so an infection does not occur.
Above explanations we included into text.
Clinical Outbreaks:
The major clinical outbreaks mentioned in the introduction would be more effective if presented in a tabular format.
A few major clinical outbreaks concerning microbial contaminated drugs are discussed in the introduction tezt, therefore we did not make a separate table, which would contain little data.However, in tables 3-5 we have included a number of information on the isolation of pathogenic microbes from drugs.
Figure 2 – Classification of Medicines:
Please clarify whether cutaneous medicines fall under the category of non-sterile medicines.
Only cutaneous medicines used for serious wounds and burns treatment, when the skin and soft tissues are damaged, must be sterile, other cutaneous preparations (most of them) for intact skin are not sterile.
Additional Figures and Tables:
I recommend that the authors include more figures and tables throughout the review to summarize key points, classifications, mechanisms, or case studies. This will enhance the overall readability and comprehension of the manuscript.
Figures were corrected and four new tables were added.
Reviewer 3 Report
Comments and Suggestions for Authors
The authors have collected and managed the reported literature under the heading of “Microbiological Contamination of Medicinal Products - Is It a Significant Problem?” is a very interesting and informative review article. The article looks reader friendly; however, I will suggest some points to improve the article;
- The introduction is too lengthy need to shortened by focusing on clear aim and objective of the review article.
- The figures and its legends need improve e.g. the Figure 2 need to improve, it contains two parts which should be labeled and explained in Figure legend. Further, in “sterile or non-sterile” part there should be a specific name in place of “for injection” and “for inhalation”.
- Is there any study which finds the role of additives in pharmaceutical development to avoid such contamination? Explain in main text of review article.
- Is there any reported study to investigate the contamination in pharmaceuticals/drugs placed at Pharmacies to sell? Explain in main text of review article.
- A very recent interesting study; https://doi.org/10.1016/j.chmed.2025.01.006, showed natural product efficiently combat against microbes, the author may cite in section 6 (Natural Pharmaceutical Products).
Author Response
The authors have collected and managed the reported literature under the heading of “Microbiological Contamination of Medicinal Products - Is It a Significant Problem?” is a very interesting and informative review article. The article looks reader friendly; however, I will suggest some points to improve the article;
- The introduction is too lengthy need to shortened by focusing on clear aim and objective of the review article.
The introduction has been shortened and a new table has been included. In this manuscript, in addition to current data, we wanted to draw attention to the historical aspect of drug-related infections, which may be incidental or epidemic in nature.
- The figures and its legends need improve e.g. the Figure 2 need to improve, it contains two parts which should be labeled and explained in Figure legend. Further, in “sterile or non-sterile” part there should be a specific name in place of “for injection” and “for inhalation”.
We introduced the figures legend as follow:
Fig 1
Part A - sources of medicinal product contamination. a number of manufacturer stages are included, starting from the quality of raw materials through the production steps up to the administration of drugs by patient.
Part B – effect of medicinal product contamination may concern changes in the pharmaceutical form of drugs and loss of their properties, as well as adverse and harmful effects on the patient using the contaminated drug.
Fig. 2
Part A – Sterile medicines are products for humans and animals free of microorganisms, because they can be pose a high risk of infection. Different categories of parenteral preparations may be distinguished: injections, infusions, concentrates or powders for injections or infusions, gels for injection, implants, intravitreal preparations.
Part B – Non-sterile medicines contain preparations used in various routes of administration including cutaneous preparations used on intact skin. Several categories of preparation for inhalation may be distinguished: preparations to be converted into vapour, preparations for nebulization, pressurised or non-pressurised metered dose preparations for inhalation, inhalation powders.
- Is there any study which finds the role of additives in pharmaceutical development to avoid such contamination? Explain in main text of review article.
The acceptance criteria for microbiological quality of non-sterile substances (active substances and additives) for pharmaceuticals are listed in the Pharmacopoeias (see references: 87, 92, 93): TAMC 103 CFU/g or CFU/mL and TYMC 102 CFU/g or CFU/mL. From the microbiological point of view, the issue of incorporation of preservatives to drugs is important. This is described in Pharmacopoeial monographs, e.g. Ph. Eur. 5.1.3. Efficacy of Antimicrobial Preservation. In our article, we focus on discussing the broad problem of microbiological contamination and its consequences for human health. Discussing the role of individual components of the drug is out of the scope of this manuscript.
- Is there any reported study to investigate the contamination in pharmaceuticals/drugs placed at Pharmacies to sell? Explain in main text of review article.
The relevant monographs of Pharmacopoeias are cited in the text of the manuscript. In Part 7 "Other aspects of medicinal Product contamination" we discuss the issue of compounded drugs prepared in pharmacies. In our manuscript we cite a number of publications related to drug contamination in pharmacies. In Part 6 "Natural Pharmaceutical Products" we specifically discuss the issue of contamination of drug of natural origin, especially products used in traditional natural medicines, ready for sale.
- A very recent interesting study; https://doi.org/10.1016/j.chmed.2025.01.006, showed natural product efficiently combat against microbes, the author may cite in section 6 (Natural Pharmaceutical Products).
It seems to us that Reviewer 3 was mistaken in suggesting citing the above interesting article. After reading it carefully, we figured that it does not contain any microbiological information including the names of microorganisms.
Round 2
Reviewer 1 Report
Comments and Suggestions for Authors
Accept in present form
Reviewer 2 Report
Comments and Suggestions for Authors
The author revised the manuscript satisfactorily.
Reviewer 3 Report
Comments and Suggestions for Authors
The authors have addressed the comments satisfactorily; however, the revised version of review article “Microbiological Contamination of Medicinal Products - Is It a Significant Problem?” contain some minor grammatical/technical errors which should be fixed before final submission.
Comments on the Quality of English LanguageMinor grammatical errors were detected